# DNA polymerase α-primase facilitates PARP inhibitor-induced fork acceleration and protects BRCA1-deficient cells against ssDNA gaps

Zuzana Machacova[1], Katarina Chroma [1], David Lukac[1], Iva Protivankova [1] & Pavel Moudry [1] ✉

PARP inhibitors (PARPi), known for their ability to induce replication gaps and accelerate replication forks, have become potent agents in anticancer therapy. However, the molecular mechanism underlying PARPi-induced fork acceleration has remained elusive. Here, we show that the first PARPi-induced effect on DNA replication is an increased replication fork rate, followed by a secondary reduction in origin activity. Through the systematic knockdown of human DNA polymerases, we identify POLA1 as mediator of PARPi-induced fork acceleration. This acceleration depends on both DNA polymerase α and primase activities. Additionally, the depletion of POLA1 increases the accumulation of replication gaps induced by PARP inhibition, sensitizing cells to PARPi. BRCA1-depleted cells are especially susceptible to the formation of replication gaps under POLA1 inhibition. Accordingly, BRCA1 deficiency sensitizes cells to POLA1 inhibition. Thus, our findings establish the POLA complex as important player in PARPi-induced fork acceleration and provide evidence that lagging strand synthesis represents a targetable vulnerability in BRCA1-deficient cells.

Eukaryotic DNA replication starts at multiple sites known as origins of replication, from which two replication forks emerge and proceed bidirectionally. The effective duplication of the eukaryotic DNA depends on the precise activation of replication origins and the seamless progression of replication forks[1]. Human DNA polymerase α-primase (hereafter referred to as POLA complex) is a four-subunit complex consisting of primase subunits PRIM1 and PRIM2, polymerase catalytic subunit POLA1 and accessory subunit POLA2[2]. POLA complex possesses a unique combination of both polymerase and primase activities, which are important for initiating DNA replication[3]. Specifically, the POLA complex synthesizes an approximately 10-nt long RNA primer, which is subsequently extended by its DNA polymerase activity, yielding a 30-nt hybrid RNA-DNA primer. Following this priming step, the POLA complex is subsequently replaced by processive polymerases ε and δ, responsible for synthesizing the leading and lagging strands, respectively. While polymerase ε synthesizes the nascent leading strand in a continuous manner, polymerase δ synthesizes the nascent lagging strand as a series of approximately 200-nt long Okazaki fragments (OF) that are further processed by nucleases FEN1 and DNA-2 and sealed by ligase LIG1[4].

DNA replication is regulated by a plethora of additional proteins, including poly (ADP-ribose) polymerase 1 (PARP1), the major protein of the PARP family, that plays multifaceted roles in various cellular processes. In response to DNA damage, one of the initial events in the signaling cascade is the recruitment of PARP1, wherein it post-translationally attaches negatively charged poly(ADP-ribose) to itself and numerous target proteins[5,6]. Understanding the involvement of PARP in DNA repair pathways laid the foundation for the development

[1]Institute of Molecular and Translational Medicine, Faculty of Medicine and Dentistry, Palacky University, Olomouc, Czech Republic.
✉e-mail: pavel.moudry@upol.cz

of PARP inhibitors (PARPi). Notably, preclinical studies and clinical trials have underscored the remarkable efficacy of PARPi, particularly in BRCA-mutated cancers, marking a significant advance over traditional chemotherapies[7].

Initially, it was postulated that PARP inhibition led to the stalling and eventual collapse of replication forks, culminating in the generation of DNA double-strand breaks[8,9]. Given that stalled replication forks require fork protection (FP), and that double-strand breaks are typically repaired via homologous replication (HR), BRCA-mutated tumors, which exhibit impaired FP and HR activities, are extremely sensitive to PARPi. However, our previous work challenged this paradigm by revealing that PARP inhibition accelerated replication fork progression[10]. This finding led to the hypothesis that accelerated replication fork progression underlies PARPi toxicity in BRCA-mutated cancer cells, ultimately resulting in the formation of DNA double-strand breaks. Building upon these observations, subsequent studies demonstrated that PARPi-induced fork acceleration is a discontinuous process, resulting in the accumulation of single-stranded breaks, nicks, and gaps[11]. Consequently, it was proposed that the aberrant formation and repair of single-stranded DNA (ssDNA) gaps were the key factors underpinning the vulnerability of cancer cells to PARPi[12], as BRCA1/2 proteins were shown to have a critical role in replication gap suppression (RGS). The origin of these ssDNA gaps has been linked to defects in both lagging strand processing and PRIMPOL-dependent repriming reactions on the leading strand. Strikingly, PARPi was found to impede the maturation of nascent DNA strands during replication[13]. PRIMPOL, functioning as a DNA primase/polymerase, facilitates initiation of de novo DNA synthesis, enabling replication fork progression under stress conditions while concurrently generating ssDNA gaps that necessitate post-replicative DNA repair[14–18].

Despite the well-established therapeutic significance of PARPi, the precise molecular mechanism underlying PARP inhibitor-induced fork acceleration remains only partially elucidated. In this study, we address this knowledge gap by revealing that the primary impact of PARPi on DNA replication is not a reduction in origin activity but rather an increase in replication fork rate. Through systematic investigations involving the depletion of all 16 catalytic subunits of human DNA polymerases, we have identified POLA1 and PRIMPOL as mediators of PARPi-induced fork acceleration. Our data point to an important role of the POLA complex in PARPi-induced fork acceleration and reveal a rationale for targeting the POLA complex in BRCA1-deficient cancer cells.

## Results

### Increased replication fork rate, not reduced origin activity, is the primary cause of PARPi on DNA replication

We showed previously that PARP inhibition accelerates the replication fork rate and reduces origin activity[10]. Since fork rate and origin activity influence each other, it is challenging to determine the primary cause of replication stress[19]. To unveil the underlying mechanism of PARPi-induced impact on DNA replication dynamics, we first asked the question of whether fork rate or origin activity is the primary cause of PARPi on DNA replication. To this end, we employed a strategy that distinguishes between the cause and consequence of replication stress developed by the Méndez laboratory[19]. Using the DNA polymerase inhibitor aphidicolin (APH) to constrain the replication fork rate, we monitored replication fork progression by DNA combing (Fig. 1a) in the presence or absence of 10 μM PARPi Olaparib for 16 h[10]. Concurrently, we assessed origin activity by quantifying the number of origins activated during the CldU pulse (first-label origins) relative to the number of total replication structures, including origins, forks, and termination events. PARPi treatment induced fork acceleration, while APH alone and in combination with PARPi reduced fork rate (Fig. 1b). As reported previously[10], PARPi also reduced origin activity as documented by the reduction of first-label origins (Fig. 1c). More importantly, PARPi-induced reduction of first-label origins was restored by

APH (Fig. 1c), indicating that PARPi effect on origin firing is a secondary response. Our data suggests that accelerated fork rate is the primary cause of PARPi on DNA replication, while reduced origin activity is the secondary response.

### Identification of DNA polymerases responsible for PARPi-induced fork acceleration

With the increased fork rate identified as the primary driver of PARPi effects on DNA replication, we focused on identifying the particular DNA polymerases involved in this process. To accomplish this, we prepared a small-scale siRNA library targeting the catalytic subunits of all 16 human DNA polymerases. Our goal was to assess the impact of individual knockdowns on PARPi-induced fork acceleration. We identified three prominent candidate hits: POLN, PRIMPOL, and POLA1, the depletion of which substantially hindered PARPi-induced fork acceleration (Fig. 2a). To rule out the possibility that the reduced fork rate in the presence of PARPi might result from a general disruption of DNA replication, we carefully validated the roles of these three candidates, both in the presence and absence of PARPi. Firstly, the knockdown of POLN resulted in a significant reduction in fork rate, also in the absence of PARPi (Fig. 2b and Supplementary Fig. 1a), indicating that the effect of POLN depletion on fork rate is not exclusive to PARPi-induced fork acceleration and reflects its general influence on DNA replication. In contrast, PRIMPOL knockdown did not affect the replication fork rate in the absence of PARPi but effectively prevented PARPi-induced fork acceleration (Fig. 2c and Supplementary Fig. 1b). These findings corroborate recent reports implicating the PRIMPOL in response to PARPi[15,20,21], thereby validating the outcomes of our screening approach. Finally, POLA1 knockdown did not affect the replication fork rate in the absence of PARPi but prevented PARPi-induced fork acceleration (Fig. 2d and Supplementary Fig. 1c). This result did not reflect cell type-specific effect nor POLA1 siRNA off-target effects, since POLA1 knockdown prevented PARPi-induced fork acceleration in U2OS, Hela and RPE1 cells (Supplementary Fig. 2a–c) by two independent siRNAs. Collectively, these data show that POLA1 mediates PARPi-induced fork acceleration.

### POLA complex is required for PARPi-induced fork acceleration

We conducted an in-depth investigation into the contribution of individual subunits within the POLA complex to the observed PARPi-induced fork acceleration. Using siRNAs targeting POLA1, POLA2, PRIM1, and PRIM2, we evaluated their impact on PARPi-induced fork acceleration through DNA combing. Notably, the depletion of any of these POLA complex subunits prevented the PARPi-induced fork acceleration (Fig. 3a). However, a careful examination of cellular lysates by Western blot analysis showed, apart from efficient downregulation of each POLA complex subunits, that depletion of POLA2 also led to the destabilization of the POLA1 subunit (Supplementary Fig. 3a).

To further explore the role of the POLA complex in PARPi-induced fork acceleration and to overcome the limitations of siRNA depletion, we turned to POLA1 chemical inhibition that does not destabilize POLA complex subunits (Supplementary Fig. 3b, c). We used the direct allosteric POLA1 inhibitor (POLAi), CD437[22], and its more potent derivative, ST1926[23]. To determine whether POLA1 activity is critical for fork acceleration, we exposed U2OS cells to PARPi and then added 1 μM ST1926 or 2 μM CD437 and subsequently analyzed fork rate using DNA combing. Importantly, inhibition of POLA1 by both ST1926 (Fig. 3b) and CD437 (Supplementary Fig. 3d) effectively prevented PARPi-induced fork acceleration. Additionally, we explored the impact of inhibiting the primase activity of the POLA complex using a selective primase inhibitor (PRIMi), ara-Adenosine-5'-triphosphate[24]. Again, PRIMi blocked PARPi-induced fork acceleration (Fig. 3c), highlighting the dependence of PARPi-induced fork acceleration on both the primase and polymerase activities of the POLA complex.

**a**

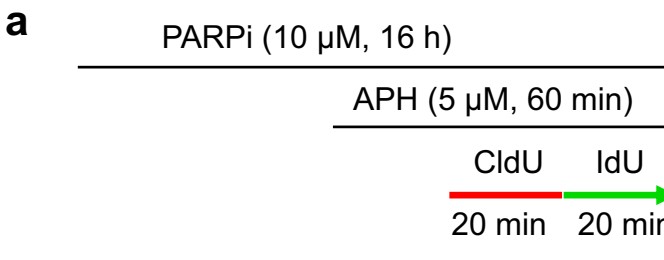

**b**

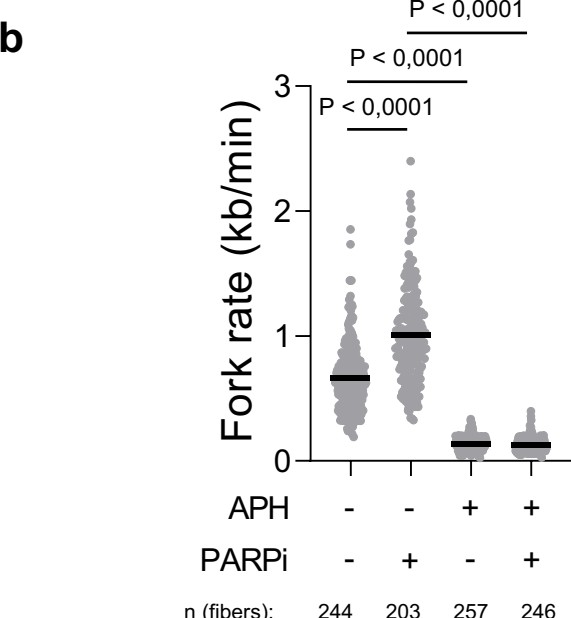

**Fig. 1 | Increased replication fork rate, not reduced origin activity, is the primary cause of PARPi on DNA replication. a** Labeling scheme to evaluate the primary effect of PARPi in U2OS cells used in **b** and **c**. **b** DNA combing assay showing that APH reduces PARPi-induced fork acceleration. The scatter plot of fork rates based on IdU tract length is presented, with the mean values marked on the graph. Each dot represents one fiber; data were from three independent experiments ($n = 3$). Statistical analysis was conducted by Kruskal–Wallis test with Dunn's multiple comparisons test. **c** Evaluation of origin activity by quantification of first-label origins showing that PARPi does not repress origin activity in the presence of APH. The mean values of three independent experiments ($n = 3$) with standard deviations indicated as error bars are shown. Statistical analysis was conducted by one-way ANOVA with Holm–Sidak's multiple comparisons test. Source data are provided as a Source Data file.

replication fork acceleration induced by TICRR or MTBP depletion (Fig. 3d and Supplementary Fig. 3e). Collectively, these findings underscore a role of both polymerase and primase activities within the POLA complex specifically in mediating PARPi-induced fork acceleration.

**POLA1 deficiency and PARPi both induce ssDNA gaps**

Given that PARPi is linked to the generation of replication-associated ssDNA gaps, which are critical for PARPi synthetic lethality[11], we sought to investigate the role of POLA1 in the formation of PARPi-induced ssDNA gaps. In our experimental conditions, we confirmed that PARPi induced the formation of S1 nuclease-sensitive ssDNA gaps (Fig. 4a). Notably, we observed the presence of ssDNA gaps in U2OS cells depleted of POLA1, both in the presence and absence of PARPi (Fig. 4a), suggesting that the POLA1 contributes to the RGS and that PARPi-induced ssDNA gaps do not require POLA1.

To corroborate these findings, we employed an alternative method to analyze ssDNA gaps based on the immunofluorescence detection of incorporated nucleoside analog CldU under non-denaturing conditions. Both PARP inhibition and POLA1 downregulation independently induced ssDNA gaps. Importantly, when POLA1 downregulation was combined with PARPi treatment, it resulted in significantly greater induction of ssDNA gaps compared to either POLA1 depletion or PARPi treatment alone (Fig. 4b). Furthermore, the depletion of POLA1 exacerbated PARPi-induced accumulation of chromatin-bound RPA32 (CB-RPA32) and γH2AX (Fig. 4c, d). To assess the functional consequences of this synergistic induction of ssDNA gaps, we investigated the sensitivity of POLA1-depleted cells to PARPi. Significantly, POLA1 downregulation led to a reduced ability to form colonies in the presence of PARPi (Fig. 4e), suggesting that POLA1 downregulation sensitizes U2OS cells to PARPi presumably by generating additional ssDNA gaps.

**BRCA1 deficiency sensitizes cells to POLA1 targeting**

The BRCA pathway plays a pivotal role in RGS, encompassing the restraint of replication fork progression during replication stress and facilitating post-replicative repair of ssDNA gaps[25–29]. Furthermore, replication gaps have been connected to PARPi sensitivity and OF processing defects in BRCA1/2-deficient cells[11,26,27,30]. Therefore, we hypothesized that inhibiting POLA1 in BRCA1-deficient cells may lead to the accumulation of replication gaps and, consequently, offer a potential therapeutic strategy. To test our hypothesis, we initially analyzed ssDNA gaps in BRCA1-deficient U2OS cells under conditions of POLA1 inhibition using an S1 nuclease assay. We observed a significant shortening of IdU tracts in BRCA1-depleted U2OS cells treated with POLAi ST1926 (Fig. 5a). This finding indicates that POLA1 inhibition generates ssDNA gaps preferentially in BRCA1-depleted cells.

Subsequently, we assessed the response of BRCA1-depleted U2OS cells to replication stress induced by POLAi by analyzing the phosphorylation of RPA32 and H2AX in whole-cell lysates through immunoblotting. Our results revealed that even mild treatment with POLAi (ST1926, 100 nM, 16 h) induced synergistic upregulation of replication

**c**

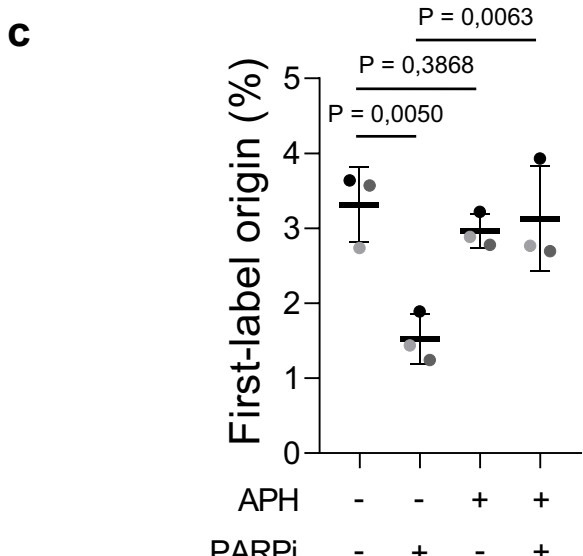

To address the concern that the general disruption of DNA replication after POLA1 depletion prevents PARPi-induced fork acceleration, we investigated whether POLA1 mediates fork acceleration induced by TICRR or MTBP downregulation[10]. Our experiments confirmed that TICRR or MTBP depletion increased the replication fork rate (Fig. 3d). More importantly, the knockdown of POLA1 did not prevent the

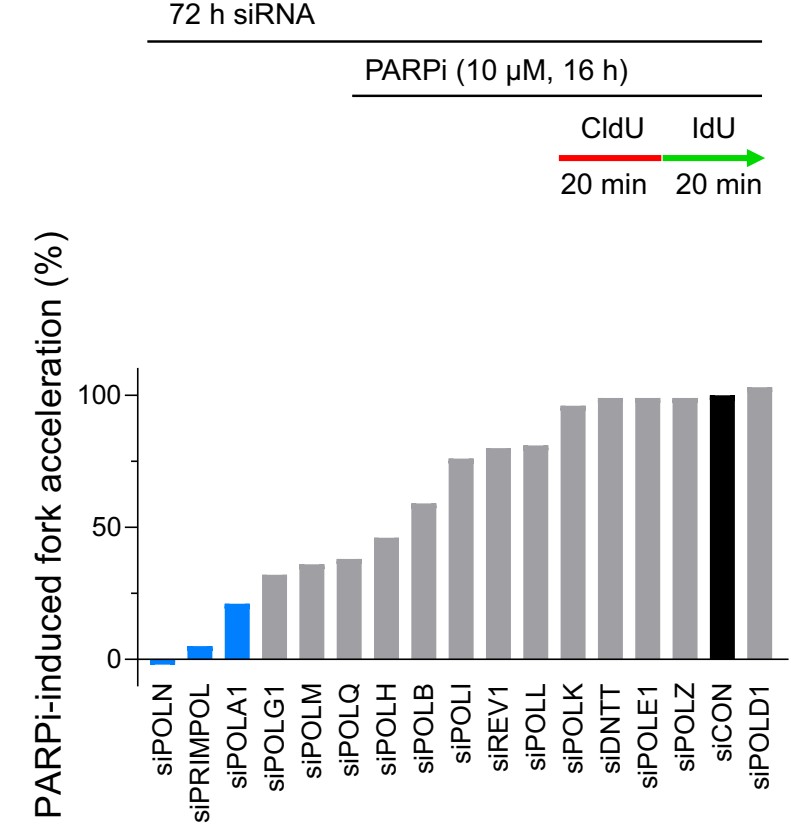

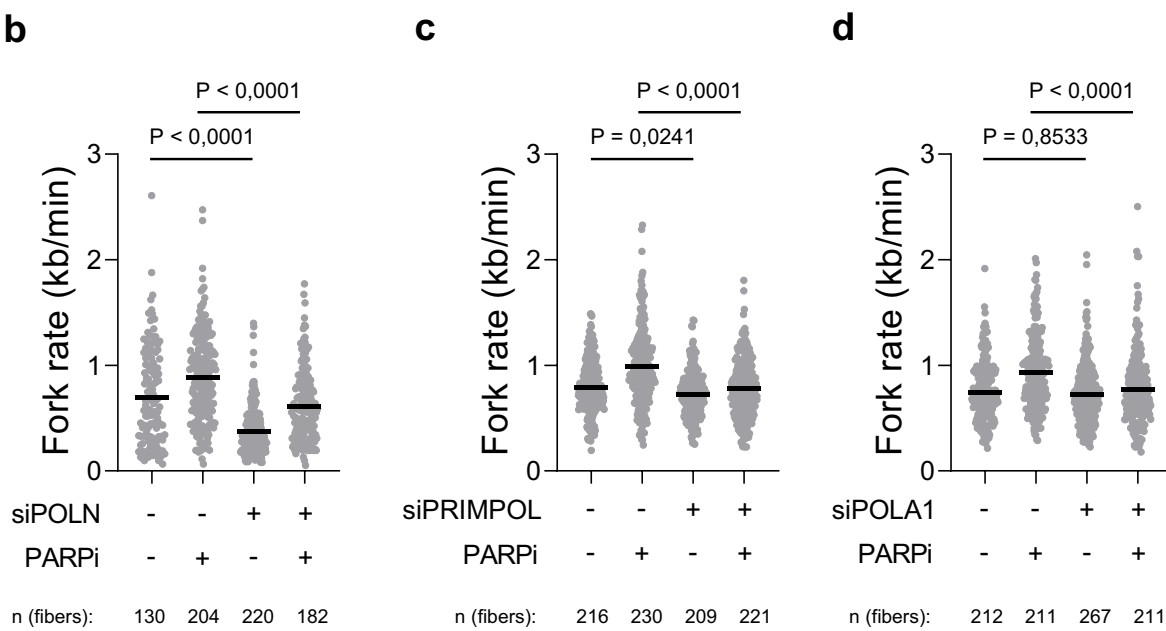

**Fig. 2 | The screen for DNA polymerases responsible for PARPi-induced fork acceleration. a** Labeling scheme for DNA combing and a bar chart showing the effects of knockdowns of individual DNA polymerases on PARPi-induced fork acceleration in U2OS cells. Data expressed are relative effects of PARPi-induced fork acceleration in cells treated with individual siRNAs relative to PARPi-induced fork acceleration in control cells (siCON = 0%, siCON + PARPi = 100%).
**b**–**d** Validation of three top-scoring hits by DNA combing analysis in U2OS cells

showing that POLN reduces replication fork rate even in the absence of PARPi, while PRIMPOL and POLA1 mediate PARPi-induced fork acceleration. The scatter plots of fork rates based on IdU tract length are presented, with the mean values marked on the graph. Each dot represents one fiber; data were from two independent experiments ($n = 2$). Statistical analyses were conducted by Kruskal–Wallis test with Dunn's multiple comparisons test. Source data are provided as a Source Data file.

**a**

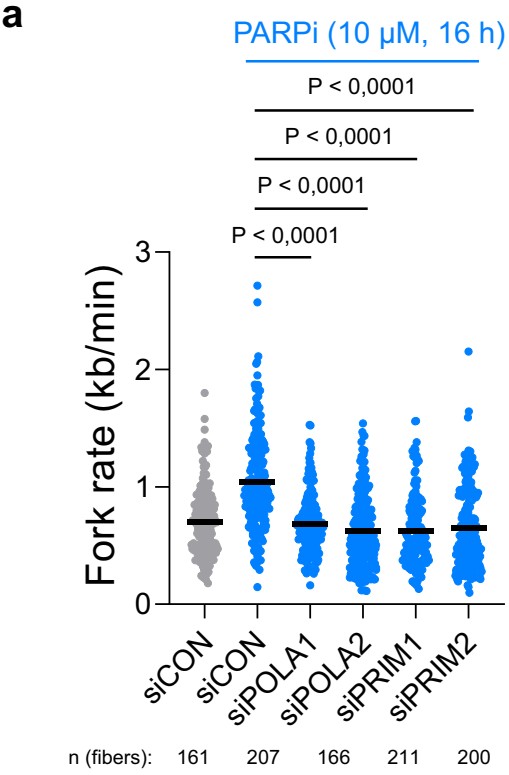

**b**

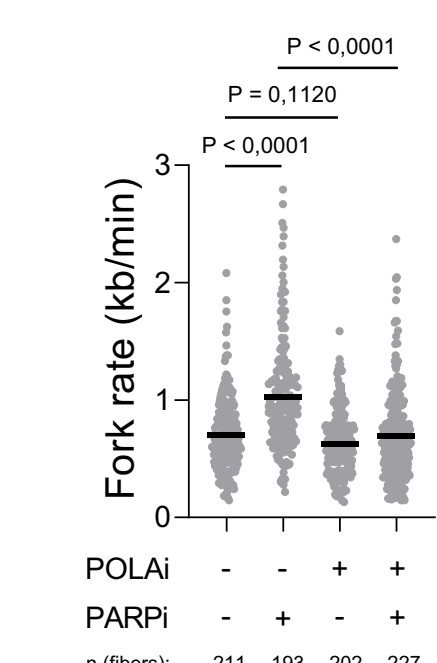

**c**

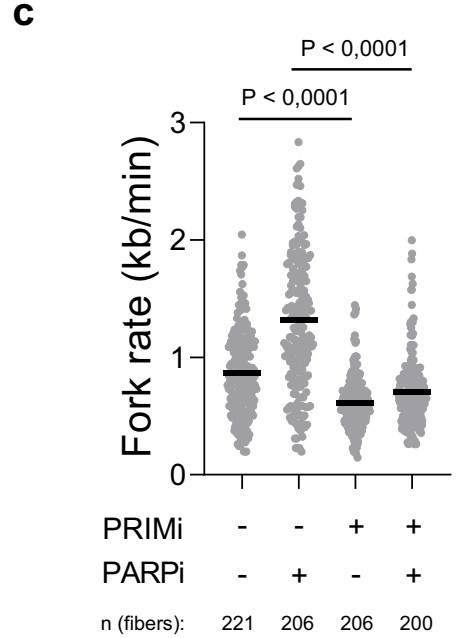

**d**

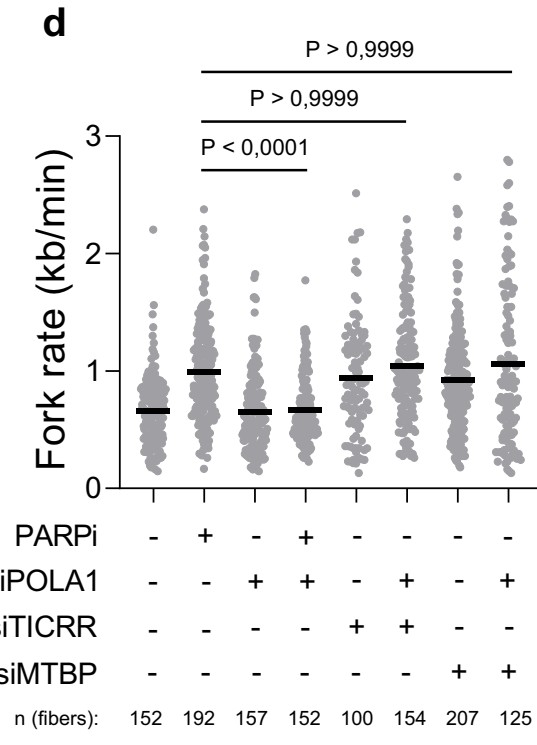

stress markers in BRCA1-depleted U2OS cells (Supplementary Fig. 4a). Additionally, short exposure to an acute dose of POLAi (ST1926, 1 μM, 20–60 min) led to a more substantial increase in RPA32 phosphorylation in BRCA1-depleted U2OS cells compared to control cells (Supplementary Fig. 4b). This finding suggests that BRCA1 plays a protective role in mitigating replication stress induced by POLAi.

Next, we asked whether the accumulation of ssDNA gaps by POLAi can be potentially exploited for targeted therapy of BRCA1-deficient cells. To address this question, we monitored the ability of both control and BRCA1-depleted U2OS cells to form colonies in the presence of POLAi. Strikingly, BRCA1-depleted U2OS cells were more sensitive to POLAi ST1926 compared to control U2OS cells (Fig. 5b). Moreover, we corroborated these data also by short-term XTT assay. Consistently, the viability of BRCA1-deficient U2OS cells was significantly compromised in the presence of POLAi ST1926 compared to control cells (Supplementary Fig. 4c).

**Fig. 3 | POLA complex is indispensable for PARPi-induced fork acceleration.**
**a** DNA combing assay showing that all POLA subunits are required for PARPi-induced fork acceleration. The scatter plot of fork rates based on IdU tract length is presented, with the mean values marked on the graph. Each dot represents one fiber; data were from three independent experiments ($n = 3$). Statistical analysis was conducted by Kruskal–Wallis test with Dunn's multiple comparisons test. **b** DNA combing assay showing that inhibition of POLA1 by 1 µM ST1926 for 20 min prevents PARPi-induced fork acceleration. The scatter plot of fork rates based on IdU tract length is presented, with the mean values marked on the graph. Each dot represents one fiber; data are from three independent experiments ($n = 3$). Statistical analysis was conducted by Kruskal–Wallis test with Dunn's multiple comparisons test. **c** DNA combing assay showing that inhibition of PRIM1 by 10 µM *ara*-ATP

for 16 h prevents PARPi-induced fork acceleration. The scatter plot of fork rates based on IdU tract length is presented, with the mean values marked on the graph. Each dot represents one fiber; data were from two independent experiments ($n = 2$). Statistical analysis was conducted by Kruskal–Wallis test with Dunn's multiple comparisons test. **d** DNA coming assay showing that POLA1 is specifically required for PARPi-induced fork acceleration and not for acceleration of replication forks induced by TICRR or MTBP knockdowns. The scatter plot of fork rates based on IdU tract length is presented, with the mean values marked on the graph. Each dot represents one fiber; data were from two independent experiments ($n = 2$). Statistical analysis was conducted by Kruskal–Wallis test with Dunn's multiple comparisons test. Source data are provided as a Source Data file.

Next, we measured POLAi sensitivity on a panel of breast and ovarian cell lines. Of note, the BRCA1-defective cancer cell lines MDA-MB-436, HCC1937, and UWB1.289 showed increased sensitivity to POLAi ST1926 relative to BRCA1-proficient cell lines (Supplementary Fig. 5a). Moreover, BRCA1-reconstituted UWB1.289 cells exhibited reduced POLAi-induced S1 nuclease-sensitive ssDNA gaps (Fig. 5c) and resistance to POLAi (Fig. 5d) compared to parental UWB1.289 cells. We noted that both parental and BRCA1-reconstituted UWB1.289 cells are still sensitive to POLAi compared to other cell lines, suggesting that additional factors modulate the sensitivity of UWB1.289 cells to POLAi. We further confirmed that POLAi-induced ssDNA gaps and cellular sensitivity to POLAi depend on BRCA1 loss using the GFP- and BRCA1-reconstituted MDA-MB-436 cancer cell lines (Fig. 5e, f). Consistently with the effects of POLAi, BRCA1 complementation in UWB1.289 and MDA-MB-436 (Supplementary Fig. 5b, d) cells prevented the formation of S1 nuclease-sensitive ssDNA gaps induced by POLA1 knockdown. Loss of ssDNA gaps in BRCA1-reconstituted UWB1.289 (Supplementary Fig. 5c) cells led to reproducible, although not statistically significant, resistance to POLA1 knockdown. BRCA1 complementation in MDA-MB-436 cells also restored resistance to POLA1 downregulation (Supplementary Fig. 5e). Overall, these findings provide a rationale for targeting POLA complex as a potential therapeutic approach for BRCA1-deficient tumors.

## Discussion

The acceleration of replication fork progression upon PARP inhibition is a well-documented phenomenon[10,11,20]. However, the molecular mechanism underlying this acceleration has remained elusive. In this study, we aimed to unravel the complexity of PARPi effects on DNA replication in human cell lines. Our findings reveal that the PARPi treatment first causes the acceleration of replication forks and is consequently accompanied by a secondary reduction in origin activity. To elucidate the mediators of PARPi-induced fork acceleration, we conducted a systematic analysis of DNA polymerases involved in this process. Our DNA combing analysis identified two contributors to PARPi-induced fork acceleration: POLA complex and PRIMPOL.

While the role of PRIMPOL in response to PARPi in unchallenged conditions has been previously reported[15,20,21], our study particularly focused on the POLA complex. Our research not only establishes the POLA complex as a player in PARPi-induced fork acceleration but also underscores the significance of DNA (re-)priming in the cellular response to PARPi. This finding is especially interesting in the context of PRIMPOL, a DNA polymerase capable of repriming de novo DNA synthesis behind replication obstacles, which has already been implicated in PARPi-induced fork acceleration.

Given the prominent role of the POLA complex in synthesizing OF on the lagging strand and the impairment of nascent DNA strand maturation by PARPi[13], it is tempting to speculate that OF processing represents a time-limiting step in DNA replication. In this scenario, PARP inhibition, by hindering OF processing, enables faster but discontinuous DNA synthesis on the lagging strand. This is consistent

with our previous observation that downregulation of OF processing enzymes LIG1 or FEN1 accelerates replication fork progression[10]. Conversely, PRIMPOL predominantly acts on the leading strand upon PARP inhibition, facilitating DNA synthesis to overcome trapped PARP1 or other replication intermediates and lesions[15,25,27–29]. While we do not exclude any potential contribution of the POLA complex to fork acceleration on the leading strand, our results best fit a scenario in which the POLA complex mediates PARPi-induced fork acceleration through its actions on the lagging strand. It is important to note that while POLA is essential for PARPi-induced replication fork acceleration, POLA is not required for the formation of ssDNA gaps by PARPi. Moreover, targeting POLA alone induces ssDNA gaps and leads to their toxic accumulation when combined with PARPi (Fig. 4f). Therefore, our findings support the model that ssDNA gaps[11], rather than accelerated replication fork rate, are critical determinants of PARPi synthetic lethality.

Our study suggests that unrestrained DNA replication under conditions of impaired OF maturation, such as PARP inhibition, poses multiple threats to genomic stability. Firstly, unligated OFs and ssDNA gaps represent DNA lesions necessitating post-replicative repair. Secondly, since DNA polymerase α lacks 3′-5′ exonuclease activity[31], and error-prone RNA-DNA primers are displaced by DNA polymerase δ during OF processing[4], abrogation of OF processing could elevate mutation rates by retaining error-prone primers synthesized by DNA polymerase α. Lastly, unprocessed RNA-DNA primers serve as a source of ribonucleotides in nascent DNA, lesions that are removed by RNase H2-dependent ribonucleotide excision repair. Notably, loss of RNase H2 sensitizes cells to PARP inhibitors[32].

Understanding the metabolism of ssDNA gaps and their cytotoxicity to cancer cells unveils new therapeutic strategies. Targeting RGS mechanisms offers precise and efficient treatments by inducing ssDNA gaps in cancer cells. It is noteworthy that the BRCA pathway also contributes to RGS[26,29], making it a promising targetable vulnerability[33]. Replication gaps in BRCA-deficient cells result from PRIMPOL-mediated repriming[15,25,29] or OF processing issues[11,34]. Our study suggests that targeting the POLA complex via maximizing ssDNA gaps sensitizes BRCA1-depleted cells and opens the possibility of uncovering additional factors that exacerbate sensitivity to POLAi. Our results are consistent with a previous study showing that BRCA2-defective cells are sensitive to POLAi[35]. Thus, our findings establish the POLA complex as a player in RGS and provide additional evidence that lagging strand synthesis represents a targetable vulnerability in BRCA-deficient cells.

## Methods
### Cell lines
Human osteosarcoma U2OS, cervical carcinoma HeLa, and diploid retinal pigment epithelium RPE1 cells were grown in DMEM (Biosera, LM-D1110/500) supplemented with 10% fetal bovine serum (Gibco, 10270106) and penicillin/streptomycin (Sigma-Aldrich, P4333). The BRCA1-defective ovarian cancer cells UWB1.289 and their complemented derivative expressing wild-type BRCA1, UWB1.289 + BRCA1,

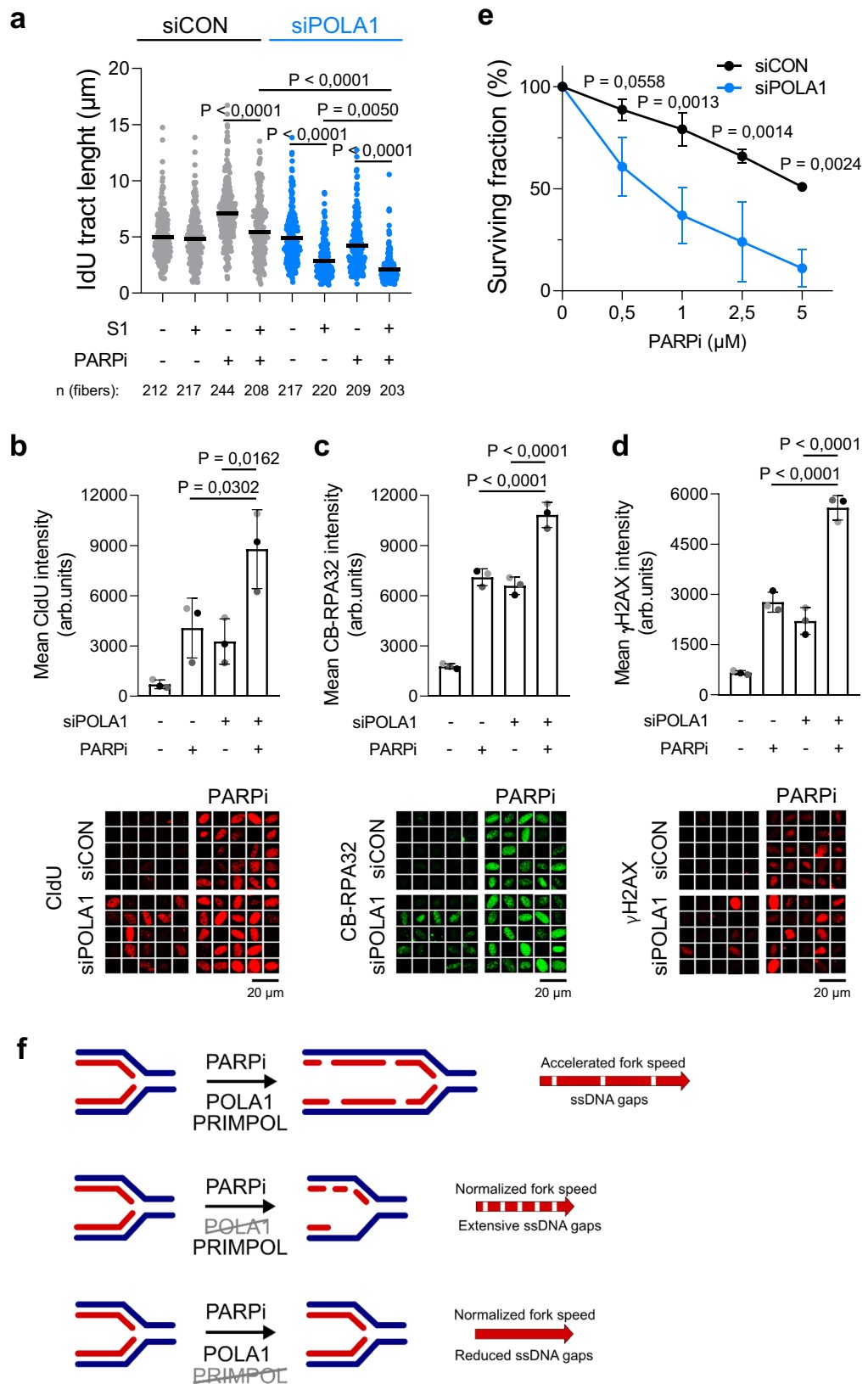

were cultivated in 50% RPMI media (Gibco, 61870), 50% MEGM BulletKit (Lonza, CC-3150) supplemented with 3% fetal bovine serum (Gibco, 10270106) and penicillin/streptomycin (Sigma-Aldrich, P4333). MDA-MB-436 cells reconstituted with GFP or BRCA1 were grown in RPMI media (Gibco, 61870) supplemented with 10% fetal bovine serum (Gibco, 10270106) and penicillin/streptomycin (Sigma-Aldrich, P4333).

U2OS (ATCC Number: HTB-96), HeLa (ATCC Number: CCL-2), RPE1 (ATCC Number: CRL-4000), UWB1.289 (ATCC Number: CRL-2945) and UWB1.289 + BRCA1 (ATCC Number: CRL-2946) cell lines were obtained from ATCC. MDA-MB-436 parental and BRCA1-reconstituted cells were kindly gifted by Neil Johnson[36]. All cell lines were regularly tested for mycoplasma contamination.

**Fig. 4 | POLA1 downregulation and PARPi both induce ssDNA gaps. a** DNA combing assay with S1 nuclease showing that POLA1 downregulation maximizes ssDNA gap accumulation in PARPi-treated U2OS cells. A scatter plot of fork IdU tract lengths is presented, with the mean values marked on the graph. Each dot represents one fiber; data were from three independent experiments ($n = 3$). Statistical analysis was conducted by Kruskal–Wallis test with Dunn's multiple comparisons test. **b** ssDNA gaps immunofluorescence experiment showing that POLA1 depletion exacerbates PARPi-induced ssDNA gap accumulation in U2OS cells. Quantifications (top) and representative images (bottom) are shown. The mean values of three independent experiments ($n = 3$) with standard deviations indicated as error bars are shown. Statistical analysis was conducted by one-way ANOVA with the Holm–Sidak multiple comparisons test. **c** Immunofluorescence experiment showing that POLA1 depletion exacerbates PARPi-induced chromatin-bound RPA32 accumulation in U2OS cells. Quantifications (top) and representative images (bottom) are shown. The mean values of three independent experiments ($n = 3$) with standard deviations indicated as error bars are shown. Statistical analysis was conducted by one-way ANOVA with the Holm–Sidak multiple comparisons test. **d** γH2AX immunofluorescence experiment showing that POLA1 depletion exacerbates PARPi-induced DNA damage accumulation in U2OS cells. Quantifications (top) and representative images (bottom) are shown. The mean values of three independent experiments ($n = 3$) with standard deviations indicated as error bars are shown. Statistical analysis was conducted by one-way ANOVA with the Holm–Sidak multiple comparisons test. **e** Clonogenic survival experiment showing that POLA1 depletion sensitizes U2OS cells to PARPi. The mean values of three independent experiments ($n = 3$) with standard deviations indicated as error bars are shown. Statistical analysis was conducted by one-way ANOVA with Tukey´s multiple comparisons test. **f** Model for the role of POLA1 in PARP inhibitor-induced replication fork acceleration. PARPi-induced fork acceleration is dependent on POLA1 and PRIMPOL. PARPi-induced ssDNA gaps are POLA1-independent resulting in extensive accumulation of ssDNA gaps and PARPi sensitivity of POLA1-depleted cells. Source data are provided as a Source Data file.

## Chemicals

In some experiments, cells were treated with the following drugs: Olaparib (Selleck Chemicals, S1060), APH (Sigma-Aldrich, A0781), ST1926 (MedChemExpress, HY-14808), CD437 (Sigma-Aldrich, C5865), ara-adenosine-5′-triphosphate (Jena Bioscience, NU-1111S).

## RNA interference

All siRNA transfections were performed using Lipofectamine RNAiMAX (Invitrogen, 13778075) according to the manufacturer's instructions. All siRNAs were obtained from Ambion as Silencer Select reagents and used at a final concentration of 14 nM. Unless specified otherwise in figure legends, most experiments were performed 48 or 72 h after transfection. siCON (negative control #1, AM4635, 5′-AGUACUGCUUACGAUACGGTT-3′), siPOLN (s51479), siPRIMPOL (s47417), siPOLA1 #1 (s10773), siPOLA1 #2 (s10772), siPOLA2 (s24281), siPRIM1(s11051), siPRIM2 (s11054), siTICRR #1 (s40362), siTICRR #2 (s40361), siTICRR #3 (s40363), siMTBP #1 (s25786), siMTBP #2 (s25787), siBRCA1 (s458) were used.

## DNA combing

Cells were labeled by sequential incorporation of 25 µM CldU (Sigma-Aldrich, I7125) and 250 µM IdU (Sigma-Aldrich, C6891) for 20 min. DNA was extracted using a FiberPrep kit (Genomic Vision, EXT-001) following the manufacturer's instructions. For experiments with the ssDNA-specific endonuclease S1, extracted DNA was incubated with or without S1 nuclease (Invitrogen, 18001-016) for 30 min at room temperature before combing on vinylsilane-coated CombiCoverslips (Genomic Vision, COV-002-RUO). Combed DNA was denatured, dehydrated, air-dried, and blocked. Coverslips were incubated with primary antibodies, mouse anti-BrdU (1:10, BD Biosciences, BD347580), and rat anti-BrdU (1:50, Abcam, ab6326) antibodies. After four washes with PBS, coverslips were incubated with secondary antibodies goat anti-mouse Alexa Fluor 488 (1:100, Invitrogen, A11001) and goat anti-rat Alexa Fluor 568 (1:100, Abcam, ab175476) antibodies. After four washes with PBS, coverslips were air-dried and mounted using Vectashield (Vector Laboratories, H-1000). Images of DNA fibers were acquired using CellObserver spinning disc confocal microscopic system (Zeiss), and the length of labeled DNA was analyzed using ImageJ software.

## qRT-PCR

Total RNA was extracted using an RNeasy mini kit (Qiagen, 74104) following the manufacturer's instructions, cDNA was generated using the RevertAid H Minus reverse transcriptase (Thermo Scientific, EP0451) and deoxynucleotide triphosphates (Promega, U120A, U121A, U122A, and U123A) and qPCR was performed using Platinum Taq DNA polymerase (Invitrogen, 15966005) and EvaGreen Dye (Biotinum, 31000) in a LightCycler Nano instrument (Roche). Relative quantity was calculated using the ΔΔCt method and GAPDH mRNA as internal normalizers. Specific primer sequences F4 and B2 for POLN were used according to a previous study[37]. F4: 5′-CAATGGACCTTTGCTCTAAACTG-3′, B2: 5′-CCGTTCTCCTGCAACAAAAT-3′.

## Immunoblotting

Cells were grown in 60 mm cell culture dishes and whole-cell extracts were obtained by lysis in Laemmli sample buffer (50 mM Tris–HCl (pH 6.8), 100 mM DTT, 2.0% SDS, 0.1% bromophenol blue, 10% glycerol) and analyzed by SDS–polyacrylamide gel electrophoresis following standard procedures. Primary antibodies were incubated overnight at 4 °C in TBS-Tween 20 containing 5% powder milk. Primary antibodies are described in Supplementary Data 1. Secondary HRP-coupled antibodies (GE Healthcare, NA931 and NA934) were incubated at room temperature for 1 h. Chemiluminescence was detected with a ChemiDoc XRS+ imaging system (Bio-Rad).

## Immunofluorescence

Cells grown on 12 mm wide glass coverslips (Karl Hecht, 41001112) were washed twice in PBS, pre-extracted in 0.5% Triton X-100 in PBS on ice for 5 min, fixed with 4% formaldehyde at RT for 15 min, washed in PBS, permeabilized for 5 min with 0.2% Triton X-100 in PBS, then washed again in PBS before being incubated with primary antibodies for 60 min at RT. Primary antibodies are described in Supplementary Data 1. After the washing step, the coverslips were incubated with goat anti-rabbit Alexa Fluor 488 or goat anti-mouse Alexa Fluor 568 secondary antibodies (Invitrogen, A11034 and A11004) for 60 min at RT, then washed with PBS, and finally mounted using Vectashield mounting medium with DAPI (Vector Laboratories, H-1200).

For the detection of ssDNA, cells were grown on coverslips in culture media with 10 µM CldU for 48 h before the indicated treatment. After treatment, cells were pre-extracted with 0.5% Triton X-100 in PBS on ice for 5 min., fixed with 4% formaldehyde at RT for 15 min, followed by immunofluorescence staining using an anti-BrdU antibody.

## Microscope image acquisition

Quantitative image-based cytometry (QIBC) of the immunofluorescence-stained samples was performed using an automatically inverted fluorescence microscope BX71 (Olympus) using scanR acquisition and analysis software (Olympus).

## Cell viability assay

Cell viability assay was performed using Cell Proliferation Kit XTT (AppliChem, A8088) according to the manufacturer's instructions. Briefly, 3000 U2OS cells per well were seeded in a 96-well plate. The next day, cells were treated with POLAi (ST1926) as indicated in figure legends. After 72 h of treatment, cells were incubated with XTT

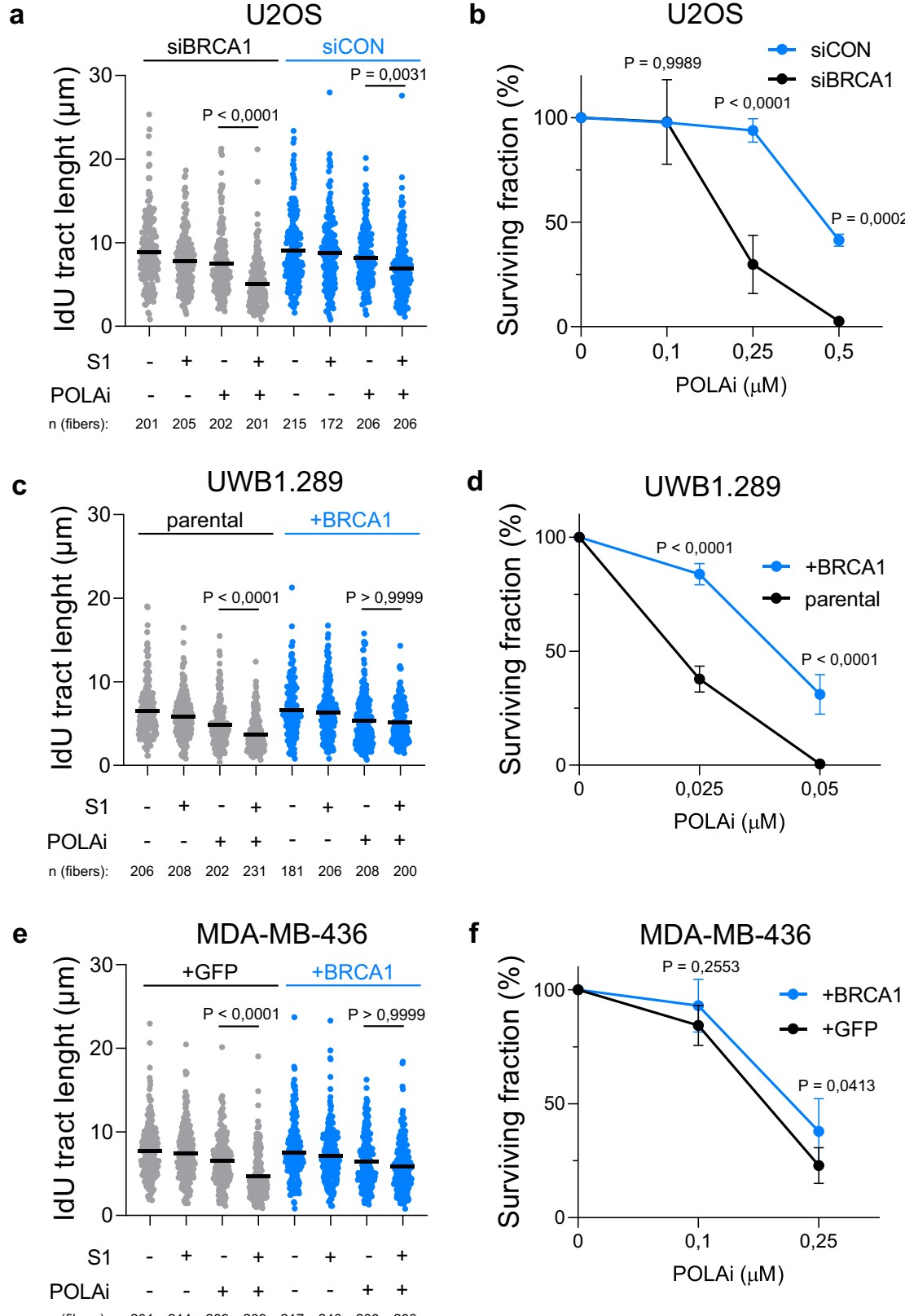

reagents, and the dye intensity was measured at 475 nm using a spectrometer (TECAN, Infinite M200PRO).

**Clonogenic assay**

Sensitivity to PARPi (Olaparib) or POLAi (ST1926) was determined by plating 100–1000 cells in a 12-well plate. The sensitivity of

UWB1.289 and MDA-MB-436 cell lines to POLA1 depletion was determined by plating 10000 cells in a six-well plate. Colonies were allowed to grow for 8–12 days, fixed in 70% ethanol, and stained with 0.5% crystal violet in 20% methanol. Colonies were counted, and the surviving fraction was calculated and normalized to untreated control.

**Fig. 5 | BRCA1 deficiency sensitizes cells to POLA1 inhibition. a** DNA combing assay with S1 nuclease showing that POLA1 inhibition (1 μM ST1926 for 20 min) maximizes ssDNA gap accumulation in BRCA1 downregulated U2OS cells. A scatter plot of fork IdU tract lengths is presented, with the mean values marked on the graph. Each dot represents one fiber; data were from three independent experiments (*n* = 3). **b** Clonogenic survival experiment showing that BRCA1 depletion sensitizes U2OS cells to POLA1 inhibitor ST1926. The mean values of three independent experiments (*n* = 3), with standard deviations indicated as error bars, are shown. **c** DNA combing assay with S1 nuclease showing that BRCA1 complementation in UWB1.289 cells prevent POLA1 inhibition (1 μM ST1926 for 20 min) induced ssDNA gap accumulation. A scatter plot of fork IdU tract lengths is presented, with the mean values marked on the graph. Each dot represents one fiber; data were from two independent experiments (*n* = 2). **d** Clonogenic survival experiment showing that BRCA1 complementation makes UWB1.289 cells resistant

to POLA1 inhibitor ST1926. The mean values of three independent experiments (*n* = 3), with standard deviations indicated as error bars, are shown. **e** DNA combing assay with S1 nuclease showing that BRCA1 complementation in MDA-MB-436 cells prevents POLA1 inhibition (1 μM ST1926 for 20 min) induced ssDNA gap accumulation. A scatter plot of fork IdU tract lengths is presented, with the mean values marked on the graph. Each dot represents one fiber; data were from two independent experiments (*n* = 2). **f** Clonogenic survival experiment showing that BRCA1 complementation makes MDA-MB-436 cells resistant to POLA1 inhibitor ST1926. The mean values of five independent experiments (*n* = 5), with standard deviations indicated as error bars, are shown. Statistical analysis was conducted by Kruskal–Wallis test, followed by Dunn's multiple comparisons test for (**a**, **c**, **e**) and one-way ANOVA with Holm–Sidak multiple comparisons test for (**b**, **d**, **f**). Source data are provided as a Source Data file.

## Reporting summary
Further information on research design is available in the Nature Portfolio Reporting Summary linked to this article.

## Data availability
All data supporting the findings of this study are available within the paper and its Supplementary Information. Source data are provided with this paper.

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

## Acknowledgements

We thank Neil Johnson for sharing MDA-MB-436 cell lines and Zdenek Skrott for critical comments on the manuscript. This project was supported by the Czech Science Foundation (grant no. 20-03457Y), the MEYS CR (Large RI Project LM2023050—Czech-BioImaging), and the project National Institute for Cancer Research (Program EXCELES, ID Project No. LX22NPO5102)—Funded by the European Union—Next Generation EU (P.M.).

## Author contributions

Z.M.: Investigation, writing—original draft, writing—reviewing and editing. K.C.: Investigation, writing— reviewing and editing. D.L.: Investigation, writing—reviewing and editing. I.P.: Investigation, writing—reviewing and editing. P.M.: Conceptualization, investigation, writing—original draft, writing—reviewing and editing, supervision, project administration, and funding acquisition.

## Competing interests

The authors declare no competing interest.
