## [Peer Review File · Nature Communications]

DNA polymerase α -primase facilitates PARP inhibition-induced fork acceleration and protects BRCA1-deficient cells against ssDNA gapsREVIEWER COMMENTS

Reviewer #1 (Remarks to the Author):

In the manuscript, “DNA polymerase α -primase facilitates PARP inhibition-induced replication fork acceleration” the authors present evidence that PARPi-induced fork acceleration is mediated by DNA polymerase α and primase activities. They also demonstrate that PolA deficiency enhances PARPi induced replication gaps. Accordingly, PolA inhibition propels gaps and sensitivity in BRCA deficient cells consistent with this background being vulnerable to the targeting of lagging strand synthesis. The manuscript is timely and has in most cases adequate experimental conditions and controls. However, as presented the manuscript does not provide clarity to what is a controversial and a complicated area, what defines PARPi sensitivity in BRCA deficient cells.

Key Points:

A major limitation of the manuscript is the lack of clarity as to what they consider fundamental to the mechanism of sensitivity upon inhibition of either PARP or PolA in BRCA deficient cells. Initially, the primary focus is on defining the cause of PARPi induced replication speed, because changes in origin activity are negated. However, it is an assumption based on the current focused analysis that fork speed is the primary driver of PARPi effects. Can a model be presented as to how PolA promotes unrestrained replication in PARPi treated cells and how does this relate to Pripol? The manuscript becomes more confusing as gaps are analyzed with respect to either PARPi or PolA loss. It is presented that the goal is to determine if PolA also contributes to PARPi induced gaps. However, as written it is not described as to why this is important. Do the authors think gaps or speed directs response? The authors state that PARPi-induced fork acceleration requires DNA polymerase α (PolA) and that PolA deficiency enhances PARPi induced replication gaps. From these two points, one would assume that fork acceleration that requires PolA is distinct from PARPi induced gaps that do not require PolA. However, this clarifying statement is not provided and again it is unclear as to the relevance of speed vs gaps.

As summarized, PARPi induces gaps are linked to lagging strand defects and PRIMPOL repriming. PolA inhibition disrupts lagging strand synthesis, but is repriming also relevant to gap formation under this condition?

It is novel that targeting PolA sensitizes BRCA deficient cells, however experiments are limited and do not consider BRCA1 mutant cancer cell lines so that relevance to cancer treatment can be considered.

Minor points:

The authors conclude that PARPi-induced effects on DNA replication is an increased replication fork rate, followed by a secondary reduction in origin activity. This outcome is expected because when replication is faster it follows that fewer dormant origins are fired. Indeed, this is what the authors find by slowing replication with APH, origin activity is recovered.

PARPi-induced fork acceleration is suggested to remain elusive, however several groups have reported on the role of PARP1 in regulating replication fork dynamics (Sugimura J Cel Biol 2008, Ray Chaudhuri NSMB 2012, and Berti NSMB 2013). Accordingly, loss of this PARP1 function interferes with replication restraint in response to replication stress.

Rather than fiber spreading the authors employ combing. This is preferred but notably the field tends towards spreading for analysis of S1 cutting. The authors should comment on their protocol

including why a 16h 10uM PARPi treatment was employed.

Reviewer #2 (Remarks to the Author):

General comments

Cancer cells carrying mutations in BRCA1 and BRCA2 genes, a condition associated with deficiency in homologous recombination-mediated DNA repair and replication fork instability, are highly sensitive to PARP inhibitors (PARPi). Sensitivity to PARPi is attributed to the role of BRCA proteins in these two processes. More recently, it has been shown that this toxicity also arises from PARPi-induced DNA replication-associated ssDNA gaps, which accelerate replication fork progression, as the authors demonstrated a few years ago.

In this manuscript, Machacova and colleagues sought to elucidate in more detail the molecular mechanism underlying the PARPi-induced acceleration of DNA replication fork progression by focusing on DNA polymerases. They depleted the catalytic subunits of all 16 human DNA polymerases in U2OS cells and verified their impact on PARPi-mediated fork acceleration. This analysis identified PRIMPOL, already known for its role in PARPi, and POLA1, which is the focus of this manuscript. The authors confirmed that the POLA complex is required for PARPi-induced fork acceleration and its inactivation, by POLA1 depletion or inhibition (by using the inhibitors ST1926 and CD437), leads to the accumulation of ssDNA gaps, which increases with siRNA-mediated BRCA1 depletion in U2OS cells. Finally, the authors demonstrated that BRCA1 deficiency (following siRNA transfection into U2OS cells) sensitizes the cells to POLA1i.

While the methodologies are mostly adequate and the experimental flow clear and consequential, the study appears rather limited in scope, and the cellular system employed is somewhat too simple to allow general conclusions to be drawn, especially regarding data on cell survival and drug response. Moreover, the originality of some of the results is mitigated by the fact that POLAi ST1926 and CD437 have already been reported by different studies to induce DNA damage and apoptosis and have demonstrated anti-tumor activity in several human tumors, as detailed below. Therefore, overall there are several weaknesses that at this point do not make it a strong candidate for publication in Nat Comms.

Major points

1. Related to the lack of originality, POLAi ST1926 and CD437 were already reported to induce early

DNA damage (by comet assay, H2AX phosphorylation and chromosomal aberrations), S-phase arrest and apoptosis in breast and colorectal cancer ((10.1097/CAD.0000000000000511; PMC5794720), in acute myeloid leukemia (AML) cell line (DOI: 10.1158/1535-7163.MCT-08-0419) and others. In this latter study the authors found that cells defective for homologous recombination (the homologous recombination (HR)-deficient and Brca2-deficient V-C8) are particularly sensitive to ST1926.

Moreover, POLA1i ST1926 was shown to reduce cell growth and to have potent anti-tumor activities in several human cancer models but not in normal counterparts: breast cancer (MCF-7, MDA-MB,231; DOI: (10.1097/CAD.0000000000000511), colorectal carcinoma (HCT116, HT29; PMC5794720), leukemia (DOI: 10.1158/1535-7163.MCT-08-0419).

Therefore, the toxic effect of POLA1i on U2OS cells reported in this study is not unexpected.

2. As already mentioned, except for the experiment in Supplementary Figure 2 (measure of fork rate by DNA fiber assay), all the data were obtained in U2OS (human osteosarcoma) cancer cell line. The authors should include in their experiments (DNA fiber data, S1 nuclease treatment, clonogenic and survival assays) additional BRCA1-deficient cells derived from cancer patients, such as MDA-MB-436, UWB1.289, SUM149PT.

3. The key functional data (e.g., in BRCA1-deficient context) should be obtained both upon depletion (siPOLA1) and inactivation (POLA1i) of POLA1.

Additional points

Figure 2b: the control for POLN depletion is missing.

Figures 3b and S3b: It would be interesting to check the effect of POLA1i on expression/stability of the other POLA components.

Figure 4a: the difference in S1-treated cells in siCTR and siPOLA1 is remarkable (around 50%), while is minor in Figure 5a. The effect of S1 nuclease treatment in condition of siBRCA1+POLA1i is high, but very similar to siPOLA1 in Figure 4a.

Figure 4e: more time points/doses are required. As for point 2, the U2OS cells are not the appropriate system and cancer-derived BRCA1 mutant cell lines should be used and compared with BRCA1 wild type cells.

The MW are missing in all Western blots and need to be added throughout the manuscript.

Page 12, lane 1: The authors mentioned that multiple siRNAs have been tested, instead only 2 were used.

Page 14, lane 22: Mistake in referring to Figure 5 (Figure 5b).

Response to Reviewers' comments

We thank all the Reviewers for their constructive comments. We have tried to address all the comments that were raised, and we believe that all the main issues have been resolved. We also provide more evidence that BRCA1 deficiency sensitizes cells to POLA inhibition.

The current version of the manuscript represents a major improvement over the original version. Figures 4e-f; 5c-f and Supplementary Figures 1a; 3b,c; 5a-e show new experiments that mostly focus on POLA1 targeting in BRCA1-deficient cells.

Overall, we hope that the Reviewers will appreciate the large number of new experiments in this revised version of the manuscript guided by their comments.

Reviewer #1 (Remarks to the Author):

In the manuscript, "DNA polymerase α -primase facilitates PARP inhibition-induced replication fork acceleration" the authors present evidence that PARPi-induced fork acceleration is mediated by DNA polymerase α and primase activities. They also demonstrate that PolA deficiency enhances PARPi induced replication gaps. Accordingly, PolA inhibition propels gaps and sensitivity in BRCA deficient cells consistent with this background being vulnerable to the targeting of lagging strand synthesis. The manuscript is timely and has in most cases adequate experimental conditions and controls. However, as presented the manuscript does not provide clarity to what is a controversial and a complicated area, what defines PARPi sensitivity in BRCA deficient cells.

We thank the Reviewer for comprehensive summary of our findings.

Key Points:

A major limitation of the manuscript is the lack of clarity as to what they consider fundamental to the mechanism of sensitivity upon inhibition of either PARP or PolA in BRCA deficient cells. Initially, the primarily focus is on defining the cause of PARPi induced replication speed, because changes in origin activity are negated. However, it is an assumption based on the current focused analysis that fork speed is the primary driver of PARPi effects. Can a model be presented as to how PolA promotes unrestrained replication in PARPi treated cells and how does this relates to Primpol? The manuscript becomes more confusing as gaps are analyzed with respect to either PARPi or PolA loss. It is presented that the goal is to determine if PolA also contributes to PARPi induced gaps. However, as written it is not described as to why this is important. Do the authors think gaps or speed directs response?

We agree with the Reviewer that the source of PARPi sensitivity in BRCA deficient cells is controversial topic in the field and our previous version of the text was not completely clear. In response to this point, we improved the results and discussion of the revised version of the manuscript, added the model of POLA-mediated replication in PARPi-treated cells (Figure 4f) and modified the title of the manuscript.

We respectfully would like to point out that experiments in Figure 1 are not designed to define the cause of PARPi sensitivity in BRCA deficient cells. These experiments, by defining the

replication fork acceleration as the primary driver of PARPi effects on DNA replication, are not meant to establish whether fork speed or ssDNA gaps define PARPi sensitivity in BRCA deficient cells. In other words, by defining that fork speed is the primary driver (and origin activity secondary) of PARPi effects on DNA replication, we do not exclude nor confirm the role of ssDNA gaps in sensitizing lesions of PARPi in BRCA-defective cells.

On the contrary, our data in Figure 4 showing that POLA1 downregulation normalizes fork speed in presence of PARPi, induces S1-nuclease sensitivity and ultimately sensitizes cells to PARPi, support the model that ssDNA gaps, not the accelerated fork speed are PARPi sensitizing lesions.

We do observe an analogous scenario in targeting POLA1 in BRCA1-deficient cells as shown in Figure 5 and Supplementary Figure 5. POLA1 targeting does not accelerate fork rate but induces ssDNA gaps in BRCA1 defective cells and sensitivity of BRCA1 defective cells. Again, this is consistent with the model that ssDNA gaps, not the fork speed, define sensitivity in BRCA1 deficient cells.

The authors state that PARPi-induced fork acceleration requires DNA polymerase α (PolA) and that PolA deficiency enhances PARPi induced replication gaps. From these two points, one would assume that fork acceleration that requires PolA is distinct from PARPi induced gaps that do not require PolA. However, this clarifying statement is not provided and again it is unclear as to the relevance of speed vs gaps.

Thank you for pointing out this important point. We modified the results and discussion of the manuscript accordingly.

As summarized, PARPi induces gaps are linked to lagging strand defects and PRIMPOL repriming. PolA inhibition disrupts lagging strand synthesis, but is repriming also relevant to gap formation under this condition?

Reviewer Fig. 1

We performed this experiment. PRIMPOL downregulation suppressed S1 nuclease sensitivity induced by POLA inhibitor ST1926 in U2OS cells (Reviewer Figure 1a), but on the other hand, PRIMPOL downregulation did not prevent accumulation of chromatin-bound RPA32 induced by POLAi in U2OS cells (Reviewer Figure 1b). We believe that this issue deserves more detailed investigation which we would like to describe in another manuscript. Therefore, we hope that the Reviewer will be satisfied by us showing the data here. Of course, if the Reviewer wants us to include these data in this manuscript, then we will do so.

It is novel that targeting PolA sensitizes BRCA deficient cells, however experiments are limited and do not consider BRCA1 mutant cancer cell lines so that relevance to cancer treatment can be considered.

Thank you for this important suggestion on how to improve the relevance of our findings to cancer treatment. We tested sensitivity to POLAi using a panel of breast and ovarian cancer cell lines. We show in new Supplementary Figure 5a, that the BRCA1-defective cancer cell lines MDA-MB-436, HCC1937, and UWB1.289 showed increased sensitivity to POLAi ST1926 relative to BRCA1-proficient cell lines HS578T, CAL51, MCF7 and MDA-MB-231.

We now provide new data using BRCA1-defective UWB1.289 cancer cell line and its reconstituted BRCA1 WT counterpart. We show in new Figures 5c and 5d that BRCA1-reconstituted UWB1.289 cells exhibited reduced POLAi-induced S1 nuclease-sensitive ssDNA gaps (Figure 5c) and resistance to POLAi (Figure 5d) compared to parental UWB1.289 cells.

We performed similar experiments using breast cancer cell line MDA-MB436 that contains the 5396 + 1G>A mutation in the splice donor site of exon 20. We show in new Figure 5e that BRCA1-reconstituted MDA-MB-436 cells exhibited reduced POLAi-induced S1 nuclease-sensitive ssDNA gaps compared to MDA-MB-436 control cells. Loss of ssDNA gaps in BRCA1-reconstituted MDA-MB-436 cells led to reproducible, although not statistically significant, resistance to POLAi (Figure 5f).

Minor points:

The authors conclude that PARPi-induced effects on DNA replication is an increased replication fork rate, followed by a secondary reduction in origin activity. This outcome is expected because when replication is faster it follows that fewer dormant origins are fired. Indeed, this is what is what the authors find by slowing replication with APH, origin activity is recovered.

We agree with the Reviewer that the previous version of the manuscript was not completely clear, we improved the text in the revised version of the manuscript. PARPi increases both fork rate and reduces origin activity, this is an observation that we and other labs found previously. Since fork rate and origin activity influence each other, there are two alternative scenarios how can PARPi affect replication dynamics.

First, PARPi increases fork rate, and this is balanced by reduced origin activity. Cause: increased fork rate, consequence: reduced origin activity.

Second, PARPi reduces origin activity, and this is compensated by an increased fork rate. Cause: reduced origin activity, consequence: increased fork rate.

Our experiments in Figure 1 are designed to answer the question of which scenario is correct. This was important at the beginning of the project to direct our focus on factors regulating fork

rate or origin firing. Slowing replication with APH in presence of PARPi restores origin activity (Figure 1) suggesting that origin activity is not primarily reduced by PARPi, instead it is secondary response to PARPi-induced fork acceleration. An example of the second scenario is treatment with CDC7 inhibitor that, like PARPi, increases both fork rate and reduces origin activity. In this case, slowing down fork rate by APH does not restore origin activity (Rodriguez-Acebes 2018, doi: 10.1074/jbc.RA118.003740), which is consistent with the role of CDC7 in origin firing.

PARPi-induced fork acceleration is suggested to remain elusive, however several groups have reported on the role of PARP1 in regulating replication fork dynamics (Sugimura J Cel Biol 2008, Ray Chaudhuri NSMB 2012, and Berti NSMB 2013). Accordingly, loss of this PARP1 function interferes with replication restraint in response to replication stress.

We kindly wish to mention the difference of “unrestrained” fork speed caused by PARP inhibition in presence of restrainer (CPT or other drugs) and our PARPi-induced accelerated fork speed (without presence of restrainer). Whereas in presence of restrainer PARPi treated cells does not slow down forks as much as restrained forks, the phenomenon of PARPi-accelerated forks refers to fork rate that is higher than in control cells.

Another point that we want to mention is that we were the first to show in our publication in 2018 (Maya-Mendoza 2018, doi: 10.1038/s41586-018-0261-5)that PARP inhibitors accelerate replication fork rate. Since it was previously accepted that PARP inhibitors lead to fork stalling, those publications do not reflect our findings from 2018 in their conclusions. If requested by the Reviewer, we are happy to cite those studies as examples of replication fork dynamics by PARP1 in response to replication stress.

Rather than fiber spreading the authors employ combing. This is preferred but notably the field tends towards spreading for analysis of S1 cutting.

We respectfully would like to point out that DNA combing leads to a parallel and uniform distribution of labeled DNA molecules. DNA combing allows detection of strand-specific alternations by resolving single chromatids, whereas DNA spreading does not (Meroni 2024, doi: 10.1083/jcb.202305082). For analysis of S1 sensitivity, DNA combing is being used in various labs in the field, including Caldecott lab (Vaitsiankova 2022, doi: 10.1038/s41594-022-00747-1), Moldovan lab (Meha 2022, doi: 10.1126/sciadv.abm0314) and Cortez lab (Hale, 2023 doi: 10.1038/s41467-023-42011-0). For these reasons, we use DNA combing rather than DNA spreading to analyze S1-nuclease sensitivity.

The authors should comment on their protocol including why a 16h 10uM PARPi treatment was employed.

For PARPi treatment, we used conditions according to our initial publication (Maya-Mendoza 2018, doi: 10.1038/s41586-018-0261-5), this is now specified in the revised version of the manuscript.

Reviewer #2 (Remarks to the Author):

General comments

Cancer cells carrying mutations in BRCA1 and BRCA2 genes, a condition associated with deficiency in homologous recombination-mediated DNA repair and replication fork instability, are highly sensitive to PARP inhibitors (PARPi). Sensitivity to PARPi is attributed to the role of BRCA proteins in these two processes. More recently, it has been shown that this toxicity also arises from PARPi-induced DNA replication-associated ssDNA gaps, which accelerate replication fork progression, as the authors demonstrated a few years ago.

In this manuscript, Machacova and colleagues sought to elucidate in more detail the molecular mechanism underlying the PARPi-induced acceleration of DNA replication fork progression by focusing on DNA polymerases. They depleted the catalytic subunits of all 16 human DNA polymerases in U2OS cells and verified their impact on PARPi-mediated fork acceleration. This analysis identified PRIMPOL, already known for its role in PARPi, and POLA1, which is the focus of this manuscript. The authors confirmed that the POLA complex is required for PARPi-induced fork acceleration and its inactivation, by POLA1 depletion or inhibition (by using the inhibitors ST1926 and CD437), leads to the accumulation of ssDNA gaps, which increases with siRNA-mediated BRCA1 depletion in U2OS cells. Finally, the authors demonstrated that BRCA1 deficiency (following siRNA transfection into U2OS cells) sensitizes the cells to POLA1i.

While the methodologies are mostly adequate and the experimental flow clear and consequential, the study appears rather limited in scope, and the cellular system employed is somewhat too simple to allow general conclusions to be drawn, especially regarding data on cell survival and drug response. Moreover, the originality of some of the results is mitigated by the fact that POLAi ST1926 and CD437 have already been reported by different studies to induce DNA damage and apoptosis and have demonstrated anti-tumor activity in several human tumors, as detailed below. Therefore, overall there are several weaknesses that at this point do not make it a strong candidate for publication in Nat Comms.

We thank the Reviewer for the extensive review of our findings.

Major points

1. Related to the lack of originality, POLAi ST1926 and CD437 were already reported to induce early DNA damage (by comet assay, H2AX phosphorylation and chromosomal aberrations), S-phase arrest and apoptosis in breast and colorectal cancer ((10.1097/CAD.0000000000000511; PMC5794720), in acute myeloid leukemia (AML) cell line (DOI: 10.1158/1535-7163.MCT-08-0419) and others. In this latter study the authors found that cells defective for homologous recombination (the homologous recombination (HR)-deficient and Brca2-deficient V-C8) are particularly sensitive to ST1926.

Moreover, POLA1i ST1926 was shown to reduce cell growth and to have potent anti-tumor activities in several human cancer models but not in normal counterparts: breast cancer (MCF-7, MDA-MB,231; DOI: (10.1097/CAD.0000000000000511), colorectal carcinoma (HCT116, HT29; PMC5794720), leukemia (DOI: 10.1158/1535-7163.MCT-08-0419).

Therefore, the toxic effect of POLAi on U2OS cells reported in this study is not unexpected.

Indeed, we agree with the Reviewer that POLA inhibitors are known to induce DNA damage and replication stress. We respectfully would like to point out that our results provide novel insights into anti-tumor activities of POLA inhibitors in following ways.

Our work focuses on response of BRCA1-defective cells to POLA inhibitors. We believe that we are first ones to show that cells defective in BRCA1 are particularly sensitive to POLA1 targeting. In addition, we explore molecular details of BRCA1-POLA1 interaction and propose single-stranded DNA gaps as novel mechanism of sensitivity. Moreover, in the revised version of the manuscript, we provide additional evidence that also patient-derived BRCA1-defective cells are sensitive to POLA1 targeting. (Figure 5, Supplementary Figures 4 and 5).

2. As already mentioned, except for the experiment in Supplementary Figure 2 (measure of fork rate by DNA fiber assay), all the data were obtained in U2OS (human osteosarcoma) cancer cell line. The authors should include in their experiments (DNA fiber data, S1 nuclease treatment, clonogenic and survival assays) additional BRCA1-deficient cells derived from cancer patients, such as MDA-MB-436, UWB1.289, SUM149PT.

As suggested by the Reviewer, we performed additional experiments using patients-derived BRCA1-defective MDA-MB-436 and UWB1.289 cells and their counterparts complemented with BRCA1 WT (Please see above in response to Reviewer no. 1).

3. The key functional data (e.g., in BRCA1-deficient context) should be obtained both upon depletion (siPOLA1) and inactivation (POLAi) of POLA1.

We performed experiments suggested by the Reviewer using BRCA1-defective UWB1.289 cancer cell line and its reconstituted BRCA1 WT counterpart. In new Supplementary Figures 5b and 5c, we show that BRCA1-reconstituted UWB1.289 cells exhibited reduced POLA1 knockdown-induced S1 nuclease-sensitive ssDNA gaps (Supplementary Figure 5b) and reduced sensitivity to POLA1 depletion (Supplementary Figure 5c) compared to parental UWB1.289 cells.

We performed similar experiments using breast cancer cell line MDA-MB436 that contains the 5396 + 1G>A mutation in the splice donor site of exon 20. We show in Supplementary Figure 5d that BRCA1-reconstituted MDA-MB-436 cells exhibited reduced POLA1 knockdown-induced S1 nuclease-sensitive ssDNA gaps compared to MDA-MB-436 control cells. Loss of ssDNA gaps in BRCA1-reconstituted MDA-MB-436 cells led to reduced sensitivity to POLA1 depletion (Supplementary Figure 5e).

For key functional experiments using inactivation of POLA1 by inhibitors please see the response to Reviewer no.1.

Additional points

Figure 2b: the control for POLN depletion is missing.

Reviewer Fig. 2

Unfortunately, we were unable to obtain a suitable antibody that can detect endogenous POLN protein. We tested 3 different anti-POLN antibodies, none of which recognizes endogenous POLN (MW around 100 kDa) in U2OS cells. We also show that none of the three POLN siRNAs that efficiently reduce POLN mRNA (Reviewer Figure 2a) diminish any band attributable to POLN (Reviewer Figure 2a). We therefore provide control for POLN depletion for the experiment in Figure 2b by analysis of remaining POLN mRNA levels using qRT-PCR in the revised version of the manuscript (Supplementary Figure 1a).

Figures 3b and S3b: It would be interesting to check the effect of POLAi on expression/stability of the other POLA components.

We performed the experiment suggested by the Reviewer and observed that POLAi ST1926 does not affect protein level of POLA complex subunits (Supplementary Figures 3b and 3c).

Figure 4a: the difference in S1-treated cells in siCTR and siPOLA1 is remarkable (around 50%), while is minor in Figure 5a. The effect of S1 nuclease treatment in condition of siBRCA1+POLAi is high, but very similar to siPOLA1 in Figure 4a.

The Reviewer probably refers to the S1 sensitivity of POLA1 knockdown in Figure 4A and POLA1 inhibition in Figure 5a (there is no siPOLA1 in Figure 5a). We agree with the Reviewer that POLA1 downregulation (3 days after siRNA transfection) generates more pronounced S1-nuclease sensitivity than acute 20 min treatment with POLAi. Prolonged treatment with POLAi further increases S1-nuclease sensitivity in our experiments.

Figure 4e: more time points/doses are required. As for point 2, the U2OS cells are not the appropriate system and cancer-derived BRCA1 mutant cell lines should be used and compared with BRCA1 wild type cells.

We performed the experiment with additional doses of PARPi as requested by the Reviewer and again observed increased sensitivity of POLA1-depleted cells to PARPi (Figure 4e).

The MW are missing in all Western blots and need to be added throughout the manuscript.
We added MW to all western blots.

Page 12, lane 1: The authors mentioned that multiple siRNAs have been tested, instead only 2 were used.

Indeed, we tested two POLA1 siRNAs in Supplementary Figure 2. We modified the sentence to avoid confusion.

Page 14, lane 22: Mistake in referring to Figure 5 (Figure 5b).

Thank you. We fixed the text.

REVIEWER COMMENTS

Reviewer #1 (Remarks to the Author):

The revised manuscript is quite thorough and covers all of the initial critique comments. I agree with the second reviewer that the paper is on the simplistic end of the spectrum, however I think it makes a valuable contribution. I therefore recommend it is accepted.

Reviewer #2 (Remarks to the Author):

This reviewer appreciates the work done by the authors during revision and the new data added in the revised manuscript regarding some controls (expression/stability of the other POLA components upon POLAi; Suppl. Fig. 3b and 3c), and additional doses of PARPi used in Fig. 4e. Data using S1 nuclease are also clear (Fig. 5c, 5e, Suppl. Fig. 5b, 5d), but the effect on the clonogenic survival assay is more problematic. UWB1.289 cells appear hypersensitive to POLAi compared to other cells, since the 0.05 μ M dose used in Fig 5d that causes high mortality UWB1.289 cells (0% of surviving fraction), leaves other BRCA1-deficient cells substantially unaffected (Fig. 5b, 5f, Suppl. 5a). Indeed, also the BRCA1-reconstituted UWB1.289 cells appear very sensitive, showing survival of around 30% at 0.05 μ M of POLAi (Fig. 5d). This suggests that there are additional mechanisms/factors (other than BRCA1-deficiency) that exacerbate the sensitivity to POLAi in these cells, rendering them unsuitable for these studies. Also, other BRCA1-proficient cells (MCF7) appear rather sensitive to POLAi (27% of surviving fraction), further weakening the notion that BRCA1-deficiency context is the main responsible of cytotoxicity of POLAi. Finally, the other cell line investigated in more detail (MDA-MB-436) in addition to UWB1.289 cells, shows no significant difference between parental and BRCA1-reconstituted counterpart, making it difficult to support the main statement of the manuscript (Fig. 5f).

On more general context, this reviewer acknowledges that this manuscript focused on the effect of ST1926 POLAi in BRCA1-defective cells and provide additional molecular details of BRCA1-POLA1 interaction and mechanism underlying (involving the ssDNA gaps). Nevertheless, the concern related to the overall novelty remains, as this previous study mentioned (DOI: 10.1158/1535-7163.MCT-08-0419, Fig. 6b) showed that ST1926 POLAi is specifically toxic in BRCA2-defective cells in comparison to the reconstituted counterpart, already suggesting similar concept on the susceptibility of HR-deficient cells to POLAi. Certainly, this previous study should be mentioned in the text.

Response to Reviewers' comments

We would like to thank the Reviewers for the careful review of our manuscript and for providing feedback in a timely manner. This has allowed us to perform the suggested experiments and other revisions and we hope that the manuscript is now acceptable for publication.

In terms of new experiments, we updated the experiment that is reported in Fig 5f and we think that the results address the point raised by Reviewer #2.

We have also made several other changes in the manuscript to address the other points raised by Referee #2.

We thank again all Reviewers for their advice and comments, which have led to a much-improved manuscript, as compared to the original version.

Reviewer #1 (Remarks to the Author):

The revised manuscript is quite thorough and covers all of the initial critique comments. I agree with the second reviewer that the paper is on the simplistic end of the spectrum, however I think it makes a valuable contribution. I therefore recommend it is accepted.

We thank the Reviewer #1 for the positive remarks.

Reviewer #2 (Remarks to the Author):

This reviewer appreciates the work done by the authors during revision and the new data added in the revised manuscript regarding some controls (expression/stability of the other POLA components upon POLAi; Suppl. Fig. 3b and 3c), and additional doses of PARPi used in Fig. 4e. Data using S1 nuclease are also clear (Fig. 5c, 5e, Suppl. Fig. 5b, 5d), but the effect on the clonogenic survival assay is more problematic.

We would like to thank the reviewer for this positive assessment and for critically reading our manuscript.

UWB1.289 cells appear hypersensitive to POLAi compared to other cells, since the 0.05 uM dose used in Fig 5d that causes high mortality UWB1.289 cells (0% of surviving fraction), leaves other BRCA1-deficient cells substantially unaffected (Fig. 5b, 5f, Suppl. 5a). Indeed, also the BRCA1-reconstituted UWB1.289 cells appear very sensitive, showing survival of around 30% at 0.05 uM of POLAi (Fig. 5d). This suggests that there are additional mechanisms/factors (other than BRCA1-deficiency) that exacerbate the sensitivity to POLAi in these cells, rendering them unsuitable for these studies.

We agree with the Reviewer #2 that UWB1.289 cells are very sensitive to POLAi. But we are not suggesting that BRCA1 is the only factor that mediates sensitivity to POLAi. We mentioned this notion in the results of revised manuscript. In general, the sensitivity of cells to drugs can be

influenced by many factors including drug target itself, pathways related to drug metabolism or drug transporters influencing drug uptake and efflux.

Also, other BRCA1-proficient cells (MCF7) appear rather sensitive to POLAi (27% of surviving fraction), further weakening the notion that BRCA1-deficiency context is the main responsible of cytotoxicity of POLAi.

Although MCF7 cell line has WT BRCA1 gene there is allelic loss and reduced expression of transcript BRCA1 (doi:10.1158/0008-5472.CAN-05-2853). This is consistent with our data and might explain why MCF-7 cells appear sensitive to POLAi among other cell lines with WT BRCA1 gene.

Finally, the other cell line investigated in more detail (MDA-MB-436) in addition to UWB1.289 cells, shows no significant difference between parental and BRCA1-reconstituted counterpart, making it difficult to support the main statement of the manuscript (Fig. 5f).

We thank the Reviewer #2 for bringing up this point. We performed additional two replicates of experiments documenting sensitivity of MDA-MB-436 cell line to POLAi. Updated Figure 5f now includes data from all 5 replicates and shows statistically significant difference between GFP- and BRCA1-reconstituted counterparts of MDA-MB-436 cells.

On more general context, this reviewer acknowledges that this manuscript focused on the effect of ST1926 POLAi in BRCA1-defective cells and provide additional molecular details of BRCA1-POLA1 interaction and mechanism underlying (involving the ssDNA gaps). Nevertheless, the concern related to the overall novelty remains, as this previous study mentioned (DOI: 10.1158/1535-7163.MCT-08-0419, Fig. 6b) showed that ST1926 POLAi is specifically toxic in BRCA2-defective cells in comparison to the reconstituted counterpart, already suggesting similar concept on the susceptibility of HR-deficient cells to POLAi. Certainly, this previous study should be mentioned in the text.

We discussed study by Valli et al. as suggested by the Reviewer #2 in discussion of current version of the manuscript.